

# Chemically distinct particle phase emissions from highly controlled pyrolysis of three wood types

Anita M. Avery[1], Mariam Fawaz[2], Leah R. Williams[1], Tami Bond[3,4], Timothy B. Onasch[5]

[1]Center for Aerosol and Cloud Chemistry, Aerodyne Research, Inc., Billerica, MA 01821, USA
[2]Department of Civil and Environmental Engineering, University of Illinois, Urbana-Champaign, Urbana, IL 61801, USA
[3]Department of Mechanical Engineering, Colorado State University, Ft Collins, CO USA
[4]Department of Civil and Environmental Engineering, Colorado State University, Ft Collins, CO 80521, USA
[5]Center for Sensor Systems and Technology, Aerodyne Research, Inc., Billerica, MA 01821, USA

*Correspondence to*: Anita Avery (aavery@aerodyne.com)

**Abstract.** Wood pyrolysis is a distinct process that precedes combustion and contributes to biomass and biofuel burning gas phase and particle phase emissions. Pyrolysis is defined as the thermochemical degradation of wood, the products of which can be released directly or undergo further reaction during gas-phase combustion. To isolate and study the processes and

emissions of pyrolysis, a custom-made reactor was used to uniformly heat small blocks of wood in a nitrogen atmosphere. Pieces of maple, Douglas fir and oak wood (maximum 155 cm$^3$) were pyrolyzed in a temperature-controlled chamber set to 400, 500, or 600 °C. Real time particle phase emissions were measured with a Soot Particle Aerosol Mass Spectrometer (SP-AMS) and correlated with simultaneous gas phase emission measurements of CO. Particle and gas emissions increased rapidly after inserting a wood sample, remained high for tens of minutes, and then dropped rapidly leaving behind char. The

particulate mass loading profiles varied with elapsed experiment time, wood type and size, and pyrolysis chamber temperature. The chemical composition of the emitted particles was organic (C, H, O), with negligible black carbon or nitrogen. The emitted particles displayed chemical signatures unique to pyrolysis and notably different from flaming or smoldering wood combustion. The most abundant fragment ions in the mass spectrum were $CO^+$ and $CHO^+$, which together made up 23% of the total aerosol mass on average, whereas $CO2^+$ accounted for less than 4%, in sharp contrast with ambient

aerosol where $CO_2^+$ is often a dominant contributor. The mass spectra also showed signatures of levoglucosan and other anhydrous sugars. The fractional contribution of m/z 60, traditionally a tracer for anhydrous sugars including levoglucosan, to total loading ($f$60) was observed to be between 0.002 and 0.039, similar to previous observations from wild and controlled wood fires. Atomic ratios of oxygen and hydrogen to carbon, O:C and H:C as calculated from AMS mass spectra, varied between 0.41-0.81 and 1.06-1.57, respectively, with individual conditions lying within a continuum of O:C and H:C

for wood's primary constituents: cellulose, hemicellulose, and lignin. This work identifies the mass spectral signatures of particle emissions directly from pyrolysis, including $f$60 and $CO^+/CO_2^+$ ratio, through controlled laboratory experiments in order to help understand the importance of pyrolysis emissions in the broader context of wild and controlled wood fires.



# 1. Introduction

## 1.1 Biomass combustion processes

Biomass burning accounts for 90% of global primary organic aerosol emissions, with important consequences for climate (Jacobson, 2014; Bond et al., 2013) and human health (Sigsgaard et al., 2015; Chen et al., 2017). The term biomass burning can refer to a wide range of combustion processes, conditions, and fuels, including wildfires, laboratory burns, and household burning for heating or cooking. In wildfires, the blend of combustion conditions (e.g., flaming and smoldering) is complex and fuels (e.g., individual wood types and duff) are variable, with a resulting variability in emissions. Biofuel burning, a subset of biomass burning, is deliberate heat generation through combustion of biomass for the purpose of personal (e.g., cooking or home heating) or large-scale heating and power generation, and conditions are more controlled. Understanding these processes is important for indoor air quality exposure (Fleming et al., 2018; Weyant et al., 2019), regional air quality (Young et al., 2016; Bressi et al., 2016), and modeling climate change (Mallet et al., 2021; Brown et al., 2021; David et al., 2021; Keywood et al., 2012; Zhang et al., 2008).

Pyrolysis, in contrast to combustion, is the thermochemical degradation of solid fuel (e.g., wood) in the absence or near absence of gas phase oxygen. Pyrolysis is a set of endothermic reactions that occur within the biomass fuel and provides products that can undergo subsequent exothermic oxidation reactions like smoldering, which occurs on or near the surface, and flaming, which occurs in the gas phase outside the fuel. The gaseous emissions can remain as gases, condense as aerosol particles, or participate in combustion if oxygen is present. Smoldering and flaming require gas phase molecular oxygen to be present and can occur simultaneously with pyrolysis. Pyrolysis is an essential step in biomass burning, generating the volatile products that undergo exothermic combustion in a self-sustaining set of processes. In this work, the term "pyrolysis emissions" is used to indicate products from the thermal breakdown of wood and "combustion emissions" is used for products where pyrolysis emissions have been followed by gas-phase reactions.

Atmospheric measurements of biomass burning in lab or field studies are important for understanding emissions from real fires. Because pyrolysis products from heated biomass may further oxidize in flaming or smoldering combustion, producing gases, including CO and $CO_2$, and potentially new hydrocarbon compounds. Thus, wood combustion products result from pyrolysis and followed by oxidation reactions, yet pyrolysis products may also be directly emitted. The description of biomass burning aerosol emissions across combustion conditions has been parameterized using the modified combustion efficiency (MCE=$\Delta CO_2/(\Delta CO+\Delta CO_2)$) (Yokelson et al., 1996; Zhou et al., 2016), for example to apportion burning conditions between flaming and smoldering phases (Akagi et al., 2011). However, as pyrolysis is a distinct, yet simultaneous, set of processes from flaming and smoldering combustion, pyrolysis-related emissions may not be well correlated with MCE (Sekimoto et al., 2018).

Progress towards modeling atmospherically relevant biomass burning emissions requires understanding the complex processes involved, including pyrolysis. Recent measurements have used positive matrix factorization (PMF) to identify volatile organic compounds (VOCs) from open fire burning that are attributed to low and high temperature pyrolysis





occurring at different times during a burn (Sekimoto et al., 2018). The authors report that two VOC profiles (i.e., PMF-derived groups of measured molecules) accounted for more than 80% of the VOC emissions from the combustion of various western US fuel types and these two VOC profiles were chemically similar across fuel sources, but chemically distinct from each other. To date, similar correlations between biomass burning particle emissions and pyrolysis have yet to be identified. In this work, we present measurements of the chemical composition of aerosol particles emitted from wood pyrolysis using a custom designed open reactor and a Soot Particle Aerosol Mass Spectrometer (SP-AMS) (Onasch et al., 2012). A related paper describes yields and product distributions of pyrolysis over a larger range of experimental conditions than described here (Fawaz et al., 2021). This work represents the first chemical characterization of particles resulting from wood pyrolysis as the primary process with implications for understanding the chemistry of biomass burning emissions in a wide range of settings.

## 1.2 Composition of ambient biomass and biofuel burning aerosol emissions

The Aerodyne Aerosol Mass Spectrometer (AMS) (DeCarlo et al., 2006; Canagaratna et al., 2007) and the SP-AMS (Onasch et al., 2012) have been used to examine the chemical composition of aerosol particles in biomass burning studies. The AMS measures non-refractory, submicron chemical species mass concentration and size distributions, and the SP-AMS provides the same, with additional detection of absorbing aerosols, namely black carbon. Previous studies range from laboratory-controlled burns (Ortega et al., 2013; Selimovic et al., 2018), to wildfires (Zhou et al., 2017; Liu et al., 2017), to personal heating stoves (Corbin et al., 2015), to trash burning and brick kilns (Goetz et al., 2018). Previous work has shown that wood-fueled combustion emissions are predominantly organic in composition (>90%).

Aerosol emissions are compositionally described using Van Krevelen diagrams of the atomic ratios of hydrogen to carbon (H:C) versus oxygen to carbon (O:C) (Heald et al., 2010). The atomic ratios are useful in describing bulk chemical composition of particulate emissions from various fuels and combustion conditions. Such diagrams have also been used to describe the evolution of emissions throughout a combustion process and further processing after emission, including secondary aerosol formation downwind of controlled and wild fires (Ortega et al., 2013).

In ambient AMS measurements, biomass burning has been identified with several key tracers including levoglucosan and potassium (Lee et al., 2010; Bhattarai et al., 2019). In the AMS detection process, levoglucosan fragments upon electron impact ionization into the characteristic ions $C_2H_4O_2^+$ at m/z 60 and $C_3H_5O_2^+$ at m/z 73 (Simoneit et al., 1999). For example, Cubison (2011) used the fraction of m/z 60 to total organic aerosol ($f$60) as a signature of primary biomass burning emissions in aircraft measurements. Cubison (2011) also summarized a range of $f$60 values from a combination of laboratory and field studies, from 0.01-0.04, and showed that regardless of initial $f$60 value, the oxidative aging process results in a decrease in m/z 60 and an increase in m/z 44 ($CO_2^+$ ion, a marker of carboxylic acids as the end point of oxidation of organic compounds). Plots of $f$44 versus $f$60 are frequently used to describe the presence of biomass burning from source to downwind receptors (Collier et al., 2016). In mobile measurements, $f$60 is used with elevated CO and $CO_2$, or potassium to indicate atmospheric reactions with emissions from biomass burning over the trajectory of a plume (Zhou et





al., 2017). Positive Matrix Factorization (PMF) applied to AMS mass spectral data (Paatero and Tapper, 1994; Ulbrich et al., 2009) has identified several typical mass spectral signatures for biomass burning organic aerosol (BBOA). For complex mixtures of ambient burning types e.g. residential burning (Xu et al., 2016) or different burning-influenced air masses (Zhou et al., 2017), PMF can identify multiple types of sources.

Residential heating or cooking stoves exhibit some similar characteristics to uncontrolled burning in m/z 60 contribution, but have some differences. Corbin (2015) described fresh primary organic matter (POM) emissions from a residential heating stove as containing primarily $CO^+$ as the most abundant fragment, followed by $CO_2^+$, with H:C of 1-1.25 and O:C of 0.5 to 1. In this study, wood was sequentially loaded into the stove, and a starting phase was categorized by the time after addition before flaming began. It was noted that this starting phase, which was characterized by high organic emissions and by a decrease in MCE, was likely pyrolysis and was higher in $CO^+$ than the flaming phase.

### 1.3 Pyrolysis

Pyrolysis is a group of endothermic decomposition reactions driven by heating biomass or biofuel materials. In industrial applications, controlled pyrolysis is carried out largely for fuel extraction including syn gas and organic molecules. Closed-system pyrolysis has been modeled and verified experimentally under many conditions (Collard and Blin, 2014; Mettler et al., 2012a; Papari and Hawboldt, 2015). Emissions are described in terms of yield and chemical purity of gas phase and condensable products (i.e., liquid products) and depend on the thermophysical properties of the biomass or biofuel and the products like char that are produced during pyrolysis (Gronli and Melaaen, 2000). Secondary chemical transformations of these products may occur within the biomass material or after emission.

Open environment pyrolysis, such as occurs during atmospheric biomass burning, is related to these controlled pyrolysis processes, although the biomass fuel and environmental conditions are more heterogeneous and the emissions include gas and particle phase products. These experiments, conducted in an oxygen-free, open reactor, represent a step between closed-system pyrolysis and atmospheric biomass burning. The emissions are still controlled by the chemical and physical properties of the wood, reactor temperature, and the heating rate. Wood has a higher thermal diffusivity and lower permeability than the solid product of pyrolysis, char (Li et al., 2021). As pyrolysis progresses through the wood sample, char formation inhibits heat transfer but increases transport and emission of products, such that the chemical processes of pyrolysis create different conditions for reactions and transformations to take place within the wood or during emissions. Fawaz et al., 2020 used thermal diffusivity and permeability to model open pyrolysis emissions in the particle and gas phases.

The products from an open-system pyrolysis experiment include solid phase (char), gases (CO, $CO_2$, and VOCs, that remain in the gas phase), and condensable products that form aerosol particles when cooled, depending on the temperature of pyrolysis, concentration of products, and subsequent dilution. These gas-phase products of open pyrolysis are



fuel for gas phase combustion or precursors for secondary organic aerosol (SOA) formation, while the particle phase products are primary organic aerosol (POA).

This work utilizes instrumentation common in atmospheric science to measure the products of controlled pyrolysis.
Its purpose is to assess how the pyrolysis process under different conditions may contribute to biomass burning emissions. A fundamental understanding of pyrolysis as a thermochemical process provides the context for understanding the emission of products that condense into the aerosol phase, and consequences for atmospheric chemistry.

## 1.4 Products of wood pyrolysis

Wood is composed primarily of cellulose, hemicellulose, and lignin, polymers whose exact makeup varies by wood species. Cellulose, which makes up approximately 60 % of wood, is a series of ~1,000 beta-linked glucose monomers, $(C_6H_{10}O_5)_n$. Hemicellulose, which makes up 15-25 % of wood, is a mixture of 5 and 6 carbon sugars with ~100 linkages per polymer (Zhou et al., 2018). Finally, lignin, which makes up 15-25 % of wood, is a polymer of three primary chemical species: sinapyl, coniferyl, and coumaryl alcohols (Gu et al., 2013). All three lignin types have an aromatic $C_{10}H_{10}O_2$
structure in common, which is the only aromatic component of wood. Other inorganic species (mostly silicon, carbonate, calcium and sodium) are a small percent of wood composition but can impact pyrolysis due to catalytic properties or alteration of heat transfer.

Products of pyrolysis are attributable to either decomposition products of wood or to reactions between individual wood components. These products may further undergo secondary reactions, either before ejection from the wood, or in the
hot gas-phase temperatures surrounding the wood. The primary components of wood decompose from the polymer matrix and are emitted at different temperatures. Hemicellulose decomposes at 200-300 °C, followed by cellulose between 300-400 °C. Lignin is emitted across a broad range of temperatures spanning the ranges of both hemicellulose and cellulose, and up to 900 °C (Yang et al., 2007). This broad range of lignin emission and especially the hotter range, complicates the analysis of lignin-associated emissions. At temperatures higher than the decomposition temperatures, cellulose and hemicellulose can
undergo reactions that form products with unsaturated bonds, observations of highly unsaturated fragments in the mass spectra may be direct emissions from lignin, or from reactions of other wood components. Thus, pyrolysis product distributions emitted from real wood differs from those emitted from individual components (Hosoya et al., 2009).

## 2. Methods

### 2.1 Pyrolysis Reactor

Experiments were carried out at the Indoor Climate Research and Training (ICRT) center at the University of Illinois Urbana-Champaign in August 2018. The pyrolysis reactor is described in detail previously (Fawaz et al., 2020,



2021). Wood pieces were suspended in the center of a cylindrical heater and surrounded by a 22 LPM flow of heated of nitrogen to ensure pyrolysis rather than combustion. The wood suspended in the reactor was attached to a balance (Sartorius

ENTRIS-6202I) for quantification of mass loss. The reactor was heated to set temperatures of 400 °C, 500 °C, or 600 °C. All wood was kiln dried with moisture contents between 7 and 9%. Pieces were cut to small (3.5 x 3.8 x 2.9 cm), medium (7 x 3.8 x 2.9 cm), or large (14 x 3.8 x 2.9 cm) sizes. Three species of wood were pyrolyzed: hard maple (*acer nigrum*), Douglas fir (*pseudotsuga menziesii*) and white oak (*quercus alba*). These were chosen as a range of wood types (maple and oak are hardwoods and Douglas fir is a softwood) from North American forested areas. Maple experiments were performed in

duplicate on only small wood at all three temperatures. Douglas fir and oak were pyrolyzed at only 500 °C and 600 °C but at all three sizes. The reactor was heated and had nitrogen flowing before the wood was inserted.

Emissions from the reactor were pulled through an exhaust duct (ID ~ 20 cm) above the reactor with a constant flow of 0.21 m3/s. All pyrolysis emissions are assumed to enter the duct and yield closure studies suggest a primary dilution factor of approximately 5 (Fawaz et al., 2021). Centerline probes were used to sample gas phase CO and $CO_2$ and aerosol

particles from the exhaust duct. The aerosol sampling line was further diluted with filtered compressed air. Aerosol loading values reported here are as sampled from the exhaust duct, only accounting for the secondary dilution. The dilution setup and measurement technique were changed between the maple and other wood samples. For maple, the dilution ratio (DR) is an estimate based on the aerosol mass loading compared with other woods. For oak and Douglas fir, the DR was measured directly. A full list of experimental conditions including DR are shown in Table 1.

**2.2 Instrumentation**

The primary instrument used in this analysis was an SP-AMS equipped with a long time of flight mass spectrometer (resolving power m/Δm = 4,000). The AMS measures non-refractory components, including organics, nitrate, sulfate, ammonium, and chloride. The addition of the laser in the SP-AMS adds the ability to detect absorbing aerosol, notably refractory black carbon, rBC. In this work, it was operated with the laser off as a standard AMS for all of the data described

in the results section. Experiments with the laser on are described in Section 2.3. Data was saved every 15 to 45 s, with mass spectrum (MS) mode for mass loading at all times and sizing via particle-time-of-flight (PToF) mode for some experiments. The size distribution of particles was well within the AMS standard lens size range ($d_{va}$ = 70-700 nm, (Liu et al., 2007). A collection efficiency of 1 was assumed for all data, as particles were generally assumed to be liquid (Dauenhauer et al., 2009; Gilardoni et al., 2016) without a significant fraction of refractory components. The lack of refractory components was

confirmed by experiments described in Section 2.3.

Data were analyzed with standard Squirrel and Pika (version 1.61F and 1.21F, respectively) analysis tools. High resolution ions up to m/z 250 were fit. Gas phase $CO_2$ was not quantified, but similar work using the same wood and reactor has shown maximum $CO_2$ of approximately 120 ppm above background (Fawaz et al., 2021). In the AMS, gas phase $CO_2$ contributes to the $CO_2^+$ signal at m/z 44. A correction is applied to AMS data and can be constant or time-dependent (Allan

et al., 2004). A constant correction was used here for two reasons. First, the aerosol sampling line had a dilution ratio more





than 100 times greater than the gas phase line so any change in $CO_2$ in the aerosol line would be only a few ppm. Secondly, an experiment with a filter upstream of the AMS showed no increase in the gas-phase $CO_2^+$ contribution during pyrolysis.

**Table 1. Summary description of experimental conditions and results of chemical properties and emission-related ratios of wood pyrolysis products. The dilution ratio (DR) for maple is estimated based on aerosol emissions at similar conditions to oak and fir, when the DRs were measured directly. Results labeled * have the listed dilution ratio applied.**

| Temp (°C) | Size | Wood | DR | OsC Avg | $CO^+/CO_2^+$ ratio | ER ($\mu g/m^3$/ppm)* | EI (g/g)* | Total loading (g)* |
|---|---|---|---|---|---|---|---|---|
| 400 | S | Maple | 50* | -0.30 ± 0.06 | 9.9 | 8300 | 0.80 | 11 |
| 400 | S | Maple | 50* | -0.28 ± 0.08 | 9.8 | 8300 | 0.80 | 10 |
| 500 | S | Maple | 50* | -0.29 ± 0.09 | 10 | 3800 | 0.48 | 8.4 |
| 500 | S | Maple | 50* | -0.27 ± 0.06 | 11 | 3800 | 0.52 | 8.2 |
| 600 | S | Maple | 50* | 0.09 ± 0.03 | 10 | 140 | 0.06 | 1.2 |
| 600 | S | Maple | 50* | 0.06 ± 0.03 | 10 | 190 | 0.07 | 1.4 |
| 500 | S | Oak | 190 | 0.01 ± 0.02 | 11 | 1600 | 0.23 | 3.4 |
| 500 | M | Oak | 211 | -0.02 ± 0.03 | 11 | 1800 | 0.29 | 7.2 |
| 500 | L | Oak | 316 | -0.18 ± 0.06 | 11 | 2500 | 0.39 | 22 |
| 600 | S | Oak | 203 | 0.13 ± 0.04 | 8.5 | 270 | 0.10 | 1.4 |
| 600 | M | Oak | 190 | 0.07 ± 0.03 | 9.7 | 380 | 0.08 | 3.3 |
| 600 | L | Oak | 203 | -0.11 ± 0.05 | 10 | 540 | 0.09 | 8.5 |
| 500 | S | Fir | 211 | -0.07 ± 0.10 | 9.7 | 8400 | 0.71 | 6.3 |
| 500 | M | Fir | 193 | -0.17 ± 0.06 | 7.2 | 3100 | 0.31 | 7.3 |
| 500 | L | Fir | 277 | -0.25 ± 0.07 | 7.2 | 5400 | 0.51 | 23 |
| 600 | S | Fir | 203 | 0.07 ± 0.06 | 7.8 | 960 | 0.19 | 2.2 |
| 600 | M | Fir | 238 | -0.12 ± 0.06 | 5.4 | 700 | 0.13 | 3.7 |
| 600 | L | Fir | 492 | -0.17 ± 0.07 | 5.2 | 1200 | 0.25 | 16 |

Gas phase CO was quantified in this work, reaching a maximum of 100 ppm during the highest temperature experiments. Similarly, to $CO_2^+$, no $CO^+$ signal was observed in the AMS during the filter experiment, so it can be assumed that all $CO^+$ signal is from the particle phase. The AMS measurement of the $CO^+$ fragment at m/z 28 is complicated by its proximity to gas phase $N_2^+$ which dominates the mass spectrum. In standard AMS quantification, $CO^+$ is not separable from $N_2^+$ and is set equal to $CO_2^+$ (Aiken et al., 2008; Canagaratna et al., 2015). The identification and quantification of $CO^+$ is only possible at very high $CO^+$ loadings, such as is observed in direct biomass burning emissions (Ortega et al., 2013), or





with increased mass resolution. Both of these conditions are met in this work and $CO^+$ is quantified and reported here. The
treatment of low signal levels at the beginning and end of an experiment is discussed in the Supplementary Information (SI),
Section 1.

Here, we limit the discussion to only detected organic aerosol, as AMS-measured inorganic species were
consistently below detection limits. The one exception was chloride in the 600 °C maple experiments, primarily in the form
of HCl at m/z 36. For O/C and H/C calculations, the improved ambient method as described in Canagaratna (2015) was used
and included the measured $CO^+$.

During sampling, the experimental start time was determined by the introduction of the wood sample into the
heated reactor and the stop time was determined by the particle counts from a condensational particle counter (TSI, CPC
3010) returning to background levels.

### 2.3 Supporting Experiments

There are no standard reference spectra for the primary components of wood in the NIST spectral database and we
found no direct measurements of these components in AMS literature sources, so a separate experiment was performed to
sample pure, unpyrolyzed cellulose. Since cellulose is insoluble in water and most solvents, and to eliminate the effect of a
solvent, the following technique for dry-aerosolizing neat cellulose was used. Aerosol was generated in argon by directing a
short burst of compressed argon at a packed bed of cellulose. The resulting plume of cellulose was sampled with the same
AMS as described above. Argon was used as the carrier gas to remove the possibility of interference between $CO^+$ and $N_2^+$.

Two additional experiments were carried out to explore the differences between these pyrolysis experiments and
previous studies of combustion. First, the SP-AMS laser was turned on during one pyrolysis experiment to determine if any
refractory components were emitted. There was no refractory black carbon (rBC), $K^+$, or other difference between this
experiment and the corresponding laser-off experiment. Second, nitrogen flow to the reactor was turned off to allow
combustion to occur in addition to pyrolysis. The SP-AMS laser was turned on for this experiment as well. This experiment
confirmed the major difference between only pyrolysis and pyrolysis plus combustion – combustion results in significant
rBC and low organic aerosol loadings.

### 2.4 Analysis methods and parameterizations

The following analytical methods are used to describe the composition of aerosols from wood pyrolysis. Due to the
high degree of fragmentation in the AMS under vaporization at 600 °C followed by electron impact ionization, molecular
characterization of individual components is not possible. Atomic O/C and H/C ratios from fragment ions in the mass
spectrum are used for comparison of bulk compositional characteristics, following previously described methods
(Canagaratna et al., 2015). The average oxidation state of carbon, $\overline{OS}_C$, further simplifies a complex mixture of compounds
into a single value, approximated by $\overline{OS}_C = 2*O/C-H/C$ (Kroll et al., 2011). These values are most frequently used to
describe chemical evolution in the atmosphere but are used here as a point of comparison between experiments.





Individual fragment ions have been linked to parent tracer molecules, especially m/z 60 for levoglucosan. Previous work has used fractional contribution of an individual ion ($f$x = x/total organics) to quantify the influence of the parent. Here, we use $f$x, where x is any ion of interest, to identify potential tracers of specific wood or pyrolysis conditions. Normalizing for total organics removes the influence of some of the physical processes that result in orders of magnitude

differences in total mass loading between experiments, allowing for examination of chemical differences. Here, an individual high resolution fragment ion is used in $f$x, not the unit mass resolution signal as in previous studies.

## 3. Results

### 3.1 Temporal profiles of emissions

In this work we describe the time-dependent emission profiles as a function of fractional experiment time in order

to convey the similarities in shape at the start and end of each experiment, regardless of the experiment length, which varied by about a factor of three. The heat flux is controlled in these open pyrolysis experiments, so the time-dependent difference in gas and particle emissions between each wood size, wood type and reactor temperature depends predominately on heat and mass transport processes within the wood, as described in detail in Fawaz (2021). Fractional experiment time normalizes these time differences. The organic mass loadings as a function of fractional experiment time are displayed in Figure 1. The

duration ranged from about 10 min at 600 °C to about 25 min at 400 °C. Only small and large sized wood are shown for clarity, and duplicate experiments (maple only, Figure 1a) are shown to demonstrate the reproducibility of the experiments.

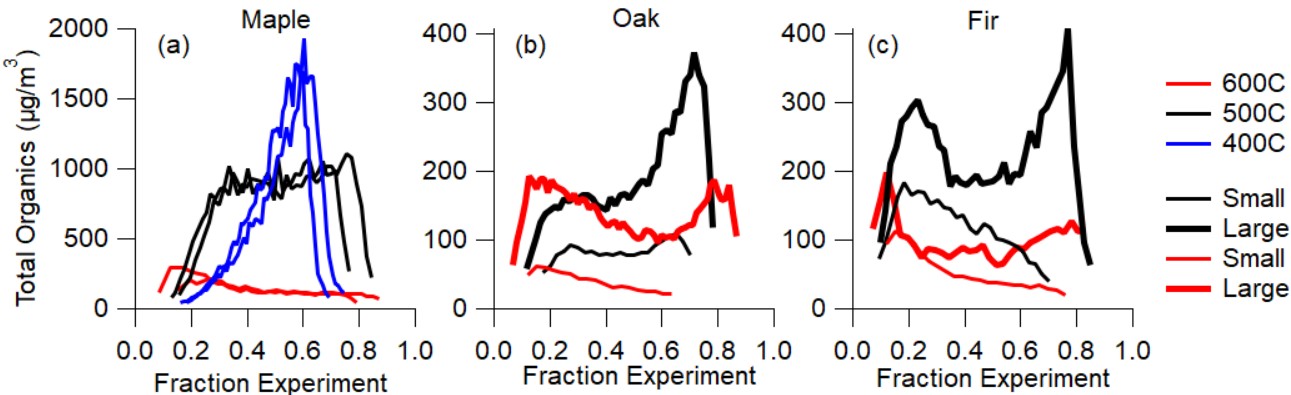

**Figure 1. Organic mass loading as a function of fractional experiment time. Note the difference in scale for the total organics**
**between maple and the other woods due to a different dilution ratio.**

The total organic loading temporal profiles are characterized by two peaks: one near the start, and one near the finish. Some experimental conditions exhibit only one peak. The first peak occurs after the initial efficient heating and





decomposition of the wood surface layers, with a reduction in heat transfer and consequent emission rate as char forms in the
surface layers and thermal conductivity is reduced. The second peak corresponds to heat reaching the center of the wood. For
small sizes, lower temperatures (400 °C experiments) exhibit emission peaks near the end, while higher temperatures (600
°C experiments) have sharp emission peaks at the beginning. Experiments at cooler reactor temperatures and larger wood
sizes produced more organic particle mass. This is reflected in the total loading values listed in Table 1.

The emission profiles for CO gas and the wood mass loss profiles exhibit similar temporal shapes as the aerosol
emissions (Fawaz et al., 2021). However, their magnitudes are reversed from the particle phase: hotter conditions produce
more CO (maximum 80 ppm for 600 °C and under 15 ppm for 400 °C) and higher mass loss rates. The CO and organic
aerosol profiles for maple at all 3 temperatures are shown in Figure S1. For all of the temporal profiles, including gas and
particle emissions and mass loss rate, reactor temperature is the primary driver, followed by wood type and size.

The high reproducibility of these temporal profiles enables estimates of open-reactor pyrolysis emission-related
enhancement ratios (ER; OA/CO ratio in µg/m$^3$/ppm) and emission indices (EI; OA/wood-mass-loss ratio in g/g) to be
determined directly from the slopes of data correlations. Table 1 lists the derived ER and EI values for each experiment. The
ER values are the fitted slope of measured organic aerosol (µg/m$^3$) times the secondary dilution ratio (DR) versus the
measured CO (ppm). The uncertainty for the ER values, including the secondary dilution values, is ~40% when DR was
measured and ~100% when DR was estimated. These laboratory-generated, pyrolysis-derived ER values represent an upper
limit of aerosol emissions from open biomass burning, as these ER values occur only when there are no subsequent gas-
phase combustion processes. The ER magnitude, like the underlying temporal profiles, is strongly dependent on the
pyrolysis temperature with the highest temperatures generating the lowest ER values. The low temperature ER values are an
order of magnitude higher than ambient biomass burning aerosol emissions; however, the high temperature ER values
overlap the range of values observed in biomass burning plume studies of 200-600 µg/m$^3$/ppm (Collier et al., 2016; de Gouw
and Jimenez, 2009).

The emission index of particle phase organic mass is presented in Figure 2d-f as the measured organic aerosol
emission rate (g/s), that is the measured OA (µg/m$^3$) times DR times flow rate through the duct (m$^3$/s), as a function of wood
mass loss rate (g/s). The slope gives mass-based EI values. The uncertainty of the EI values is greater than the ER values due
to the assumption that all emissions enter the exhaust duct.

The time-dependent oxidation state profiles of the emitted organic aerosol are shown in Figure 2a-c. The oxidation
state of emitted OA remains largely constant over the course of a pyrolysis experiment. There is some evidence of time-
dependent changes in oxidation state of emitted aerosols during the first and last 10% of the experiment, but the general
shape of change in oxidation state is independent of the total loading profile in Figure 1a-c even for fir, which exhibits the
most change in composition. For maple and oak, the oxidation state stays the same or decreases slightly over time.

The consistency in time-dependent oxidation state (OSc) is in apparent contrast with the mass loading profiles
which exhibited temporal peaks. This apparent dissonance is supported by the consistency of emissions as measured by other
parameters. At a given reactor temperature the emission rates for OA and CO and wood mass loss rates are highly correlated





and reproducible, indicating the same thermophysical processes are driving the emissions and the thermal decomposition

reactions (Fawaz et al., 2021).


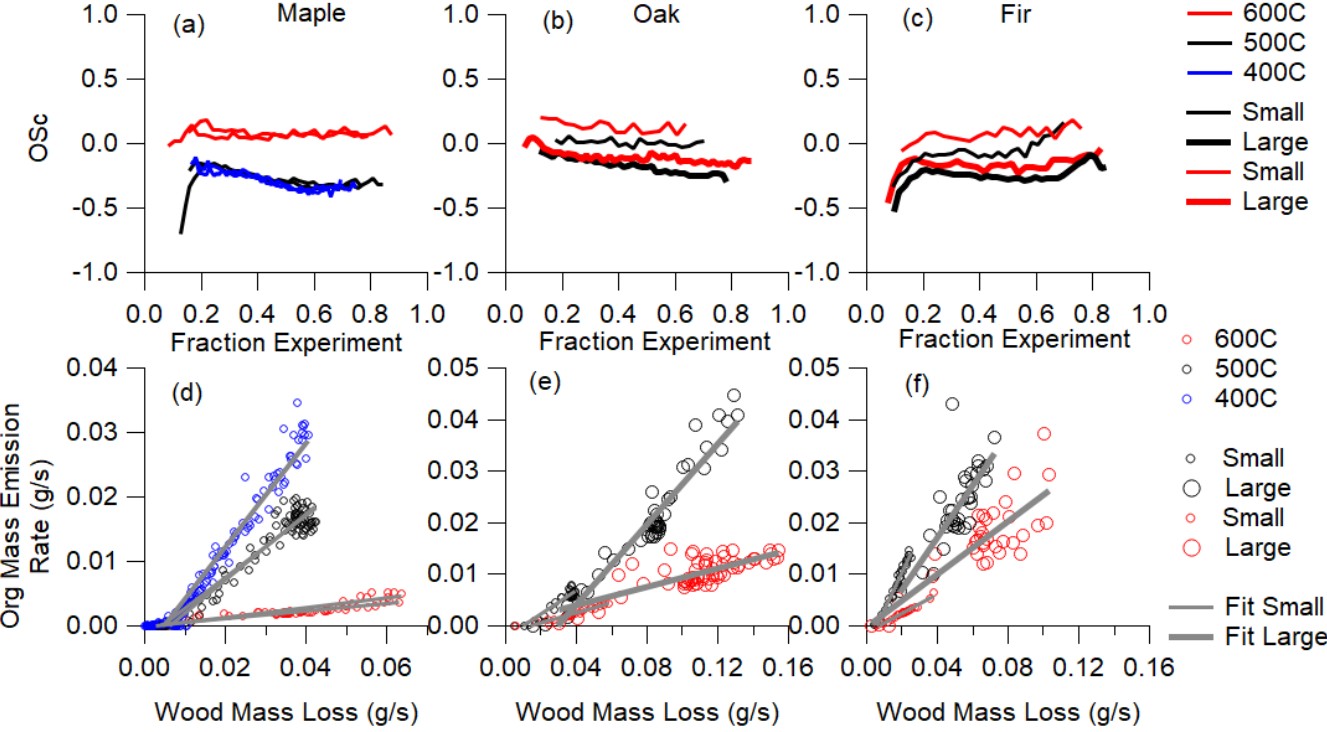

**Figure 2. The average oxidation state of condensed aerosol over the course of an experiment (a-c), and the emission rate of aerosol compared with the mass loss rate of the wood (emission index, d-f) for each experimental condition.**

305       Despite being nearly independent of time during each individual experiment, the emitted OA oxidation states do

differ with reactor temperature and wood size for a given wood type, with the exception of 400 °C and 500 °C maple.

Measured OSc increased with increasing reactor temperature, for a given wood size and type, and with decreasing wood

size, for a given reaction temperature and wood type. The change in emitted OA OSc with temperature could be due to

changes in decomposition and secondary reactions within the wood, as hinted by the changing gas phase emissions (Fawaz

et al., 2021) and temperature-dependent VOC profiles (Sekimoto et al., 2018). The change in OA OSc could also be caused

by a change in partitioning, because higher pyrolysis reactor temperatures emit lower OA levels, so less-oxidized OA could

evaporate due to its higher saturation vapor pressures (Donahue et al., 2006; Jimenez et al., 2009; Kroll et al., 2011). To

explain differences as a result of wood size, we note that the wood length that increases with size used here is with the grain

such that the channels by which the primary products leave the wood are longer. Fawaz et al. (2021) attributed a reduction in

emission quantity with wood size to transport resistance within the wood matrix, a trend that became more pronounced with





increasing wood density. Transport resistance associated with longer path length between decomposition and release could affect the chemical composition of resulting aerosols.

## 3.2 Pyrolysis aerosol chemistry

To broadly describe the chemical composition, pyrolysis emissions are summarized in Figure 3 as a Van Krevelen
diagram of H/C versus O/C. The pyrolysis results are given as blue, black, and red symbols for 400 °C, 500 °C, and 600 °C experiments, respectively. An AMS measurement of dry-generated pure cellulose is included as a green hourglass. A laboratory burn of Douglas fir carried out in 2016 during the FIREX campaign (Burn 80) in which an unknown quantity of products were consumed in gas-phase flaming combustion, is included for comparison as a green diamond. For the AMS measurements, H/C and O/C values are based on high resolution fragments, for which the uncertainties are discussed
elsewhere (Canagaratna et al., 2015). Calculated values for the primary components of wood (cellulose, hemicellulose, and lignin), based on the bulk molecular formula, are shown in green letters. It is noted that cellulose is included in Figure 3 both by formula (C) and by AMS sampling (hourglass), and that levoglucosan has the same molecular formula as cellulose. Glucose, the monomer of cellulose, is included but is quite different from the pyrolysis emissions. Hemicellulose and lignin are complex polymers without a standard molecular formula. In Figure 3, the individual lignin components, coumaryl,
coniferol, and sinapyl alcohol, are shown and have not been converted to polymer form. Since lignin has longer carbon chains, polymerization has the effect of lowering both O/C and H/C by ~0.1.

Previous pyrolysis studies of the primary components of wood show that hemicellulose and cellulose decompose at or below the temperatures of 400-600 °C used here (Yang et al., 2007). Thus, it is reasonable to assume that both hemicellulose and cellulose decompose throughout each experiment, as sufficient heat to decompose these components
continuously reaches new parts of the wood. The decomposition of lignin is not uniform and does not occur over a narrow range of temperatures, which may be the result of the nonuniform structure of lignin. (Yang et al., 2007) observed lignin decomposition to occur between 160 and 900 °C.

As Figure 3 indicates, the pyrolysis emissions observed in this work lie on a continuum between the primary components of wood, especially at lower reactor temperatures in maple and oak. Cellulose decomposition products appear to
lose oxygen and hydrogen before emission. This indicates that lower reactor temperatures result in a direct emission of primary decomposition components without secondary reaction. Higher temperatures generally result in lower H/C ratios, potentially due to reactions of primary components within the wood such as PAH formation which only occurs at higher temperatures (Lu et al., 2009). This shift could also be a result of increased lignin emission. For Douglas fir and oak, smaller wood sizes show slightly higher O/C values. This could give an indication of the effect of surface or primary emissions (due
to differences in surface area to volume ratio of the wood) with fewer secondary reactions, but a larger range of sizes would need to be analyzed to understand this potential effect.





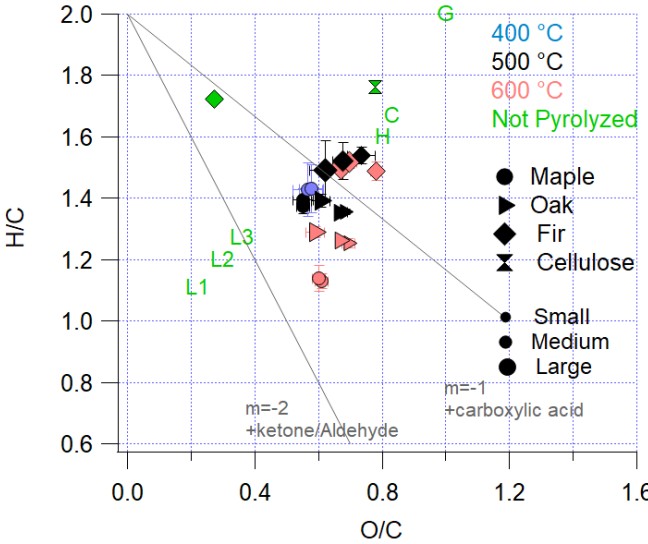

**Figure 3. Van Krevelen diagram of the pyrolysis products measured in this work. Shapes correspond to a wood type, while color corresponds to pyrolysis temperature (blue, red, black for 400 ºC, 500 ºC, and 600 ºC, respectively, and green for non-pyrolyzed materials). Note that for Fir the not pyrolyzed green diamond is burning Fir, and for cellulose (green hourglass), not pyrolyzed is dry-generated. G=Glucose C=Cellulose/Levoglucosan, H=Hemicellulose, and L1, L2, and L3 are the three primary components of lignin, coumaryl, coniferol, and sinapyl alcohols, respectively. Gray lines correspond to aging trends from fresh biomass burning, via carboxylic acid or ketone/aldehyde addition. All values are experiment averages, with bars showing one standard deviation, although this is often less than the size of the marker.**

For the hardwoods, maple and oak (apparent density 750±17 and 860±27 kg/m$^3$, respectively), the temperature increase from 500 to 600 °C results in a much lower H/C ratio, and other measurements have shown an increase in gas-phase products across this temperature range (Fawaz et al., 2021). Such a decrease in H/C is not observed for the increase from 400 °C to 500 °C for maple, or with temperature changes for Douglas fir (density 430 kg/m$^3$). This underscores the complex nature of pyrolysis as a series of temperature-dependent emission and decomposition mechanisms, complicated by both endothermic and exothermic reactions, and governed by heat transfer and mass transport (Grønli et al., 2002). However, the changes with temperature in Figure 3 are small compared to what might be observed between fresh and aged biomass burning emissions. Whether this change in H/C for different reactor temperatures in maple and oak is due to increased lignin emission and secondary reactions is unknown. If it is due to differences in lignin emission, it could explain why there is not a difference in H/C ratio between 500 °C and 600 °C for the softwood Douglas fir.

### 3.3 Pyrolysis aerosol mass spectra

Since oxidation state (Figure 2a-c) does not change much over the course of an experiment, we compare experiment-averaged aerosol mass spectra for 400 °C maple and 600 °C Douglas fir in Figure 4, along with dry-aerosolized





pure cellulose and a laboratory burn of Douglas fir. Maple and Douglas fir were chosen for comparison as they are the most disparate.

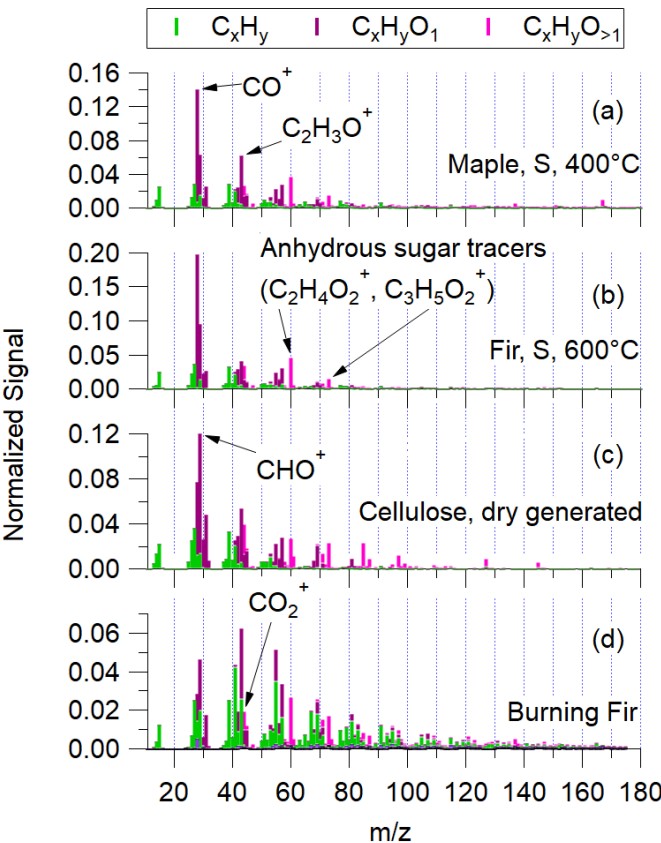

**Figure 4. The experiment-averaged mass spectra for small maple at 400 ºC (a) and small fir at 600 ºC (b) pyrolysis, as well as dry generated cellulose (c) and Douglas fir burning (d).**

Several mass spectral characteristics differ between these pyrolysis experiments and wood combustion. The first is that $CO^+$ (m/z 28) is the largest fractional contribution for every experiment by at least a factor of two. This is in contrast with aged wood combustion emissions where $CO_2^+$ (m/z 28), a marker of carboxylic acids that are ubiquitous in the atmosphere, is often the dominant ion. $CO_2^+$ is a factor of 5-8 lower than $CO^+$ in this work. The unique aspect of pyrolysis emitted aerosol mass spectra is the high $f CO^+$ signal which is not seen in atmospheric aerosols of other types (Ng et al., 2011). The next most abundant fragment ions in each spectrum are $C_2H_3O^+$ (m/z 43) or $CHO^+$ (m/z 29). All pyrolysis spectra contain notable fractions of the anhydrosugar tracer ions $C_2H_4O_2^+$ and $C_3H_5O_2^+$ (m/z 60 and 73, respectively), whereas the cellulose spectra contains these and other, larger sugar-related fragment ions.

The pure unpyrolyzed cellulose in Figure 4c suggests the source of these unique spectral characteristics. $CHO^+$, $CO^+$, and $C_2H_3O^+$ are the three most abundant fragment ions, respectively, in the cellulose aerosol mass spectra. These three





fragment ions also dominate the pyrolysis mass spectra, consistent with the products of pyrolysis being sugar-like hydrocarbons that produce small, oxygenated hydrocarbon fragments. The presence of these small fragment ions indicates that cellulose decomposition products are likely responsible for the high $f$CO$^+$ in products of pyrolyzed wood. On the other hand, CHO$^+$ is the largest contributing fragment for unpyrolyzed cellulose while CO$^+$ is largest for the pyrolysis products. This could potentially be explained by the polymerization characteristics and intra-polymer bond types of cellulose and hemicellulose as pure components instead of in real wood. Small changes to the chemical structure as a result of polymerization can have a large effect on the mass spectral signature. For example, sucrose, a polymer of glucose and fructose is characterized by many small m/z fragments, including m/z 28, that are not represented in either monomer (WebBook, 2018). In addition, heating during pyrolysis complicates this picture. The exact mechanism of emission of specific species during pyrolysis is complex, and from a molecular perspective, the depolymerization of original wood components (cellulose, hemicellulose, and lignin) could take place during decomposition, secondary reaction, or during fragmentation in the AMS.

All of the aerosol mass spectra in Figure 4, including the pyrolysis experiments, pure cellulose, and burning Douglas fir, exhibit an abundance of ion signal at m/z 60 and m/z 73, corresponding to the anhydrosugar (e.g., levoglucosan) fragments at C$_2$H$_4$O$_2^+$ and C$_3$H$_5$O$_2^+$, respectively. Emission of levoglucosan from wood during pyrolysis has been studied extensively (Shafizadeh et al., 1979; Bai et al., 2013) with yields from pure cellulose pyrolysis of 5-80% (Maduskar et al., 2018). These fragment ions are also widely used as tracers for biomass burning in ambient air (Cubison et al., 2011; Nielsen et al., 2017).

The mass spectrum of pure cellulose in Figure 4c also includes several large, oxygenated ion fragments above m/z 80, which are absent in the pyrolysis spectra. This could indicate a decomposition process that occurs during pyrolysis before further fragmentation in the vaporization and ionization processes of the AMS.

Finally, negligible nitrogen or nitrate-containing ions were observed in the aerosol mass spectra obtained in this study. Cellulose contains negligible nitrogen content. Most nitrogen content in woody plants are concentrated in the foliage, roots, and bark. Thus, debarked, dried wood samples, such as used in this study, will likely contain little nitrogen, as observed in the aerosol mass spectra for the pyrolysis experiments. In contrast, the pyrolysis-related VOC compositions observed using biomass burning experiments with more realistic mixtures of wood, bark, and foliage, indicated a strong increase in nitrogen containing compounds with higher temperatures (Sekimoto et al., 2018).

### 3.4 Individual Fragment ions and Tracers

This section discusses specific mass spectral fragment ions observed as a fraction of the total organic signal, $f$x=x/(total organics) for all of the pyrolysis experiments, as well as burning fir and cellulose. Figure 5a shows the fractional contribution ($f$x) of the small, oxidized fragment CO$^+$ (m/z 28). Median values of $f$CO$^+$ ranged from 0.12 to 0.25 across experiments (Table S1). For maple and oak, $f$CO$^+$ increased with reactor temperature, but for fir, the opposite is true at medium and large sizes. An explanation for the increase in $f$CO$^+$ with increasing temperature could be that additional





decomposition or secondary reactions occur at higher temperatures to form lower molecular weight products, but these products such as glycoaldehyde, which is a known product of cellulose pyrolysis (Richards, 1987), are too volatile to be observed in the condensed phase. Both oak and Douglas fir have a considerable wood size dependence, while maple does not. This difference between woods could be due to density or other structural differences in the wood, combined with inhibition of mass transfer caused by density. For burning fir, $f\mathrm{CO}^+$ is very low.

$f\mathrm{CO_2}^+$, shown in Figure S2c, is notable in its small magnitude, <4% of total organics. The dominance of the $\mathrm{CO}^+$ fragment and the scarcity of $\mathrm{CO_2}^+$ are both unusual for typical ambient measurements. Figure 6 shows the ratio of $\mathrm{CO}^+$ to $\mathrm{CO_2}^+$ for each experimental condition. To our knowledge, $\mathrm{CO}^+/\mathrm{CO_2}^+$ ratios above 4 have not been reported. One study in fuel-rich conditions of a personal heating stove showed $\mathrm{CO}^+/\mathrm{CO_2}^+$ ratios of nearly 4 (Corbin et al., 2015). Therefore, we recommend that $\mathrm{CO}^+/\mathrm{CO_2}^+$ greater than five be a marker for pyrolysis.

The next two most common fragments are $\mathrm{CHO}^+$ (m/z 29, Figure 5b) or $\mathrm{C_2H_3O}^+$ (m/z 43, Figure S2a); $\mathrm{CHO}^+$ is a larger contributor in Douglas fir and oak, while $f\mathrm{C_2H_3O}^+$ is larger in maple at 400 °C and 500 °C. Median values of $f\mathrm{CHO}^+$ range from 0.03 to 0.08 (Table S1) and median values of $f\mathrm{C_2H_3O}^+$ range from 0.02 to 0.05 (Table S2). $f\mathrm{CHO}^+$ is abundant in Douglas fir and exhibits inverse behavior with temperature as maple. Wood-specific and temperature-specific fragments are frequent, but it is unusual for temperature trends to be opposite between wood types, as it is for $\mathrm{CHO}^+$. This indicates that

$\mathrm{CHO}^+$ is associated with many different components or conditions. For maple, 400 °C and 600 °C are one sided in their distribution with whiskers pointed toward the value for 500°C, which has a narrower distribution, indicating a temperature dependence specific to a narrow range of compounds. A similar trend is observed for furfural-related fragments $\mathrm{C_5H_3O_2}^+$ and $\mathrm{C_5H_4O_2}^+$ (m/z 95 and 96, not shown), which may link this fragment closer to hemicellulose than cellulose (Zhou et al., 2018).

The levoglucosan tracer m/z 60 ($\mathrm{C_2H_4O_2}^+$), shown in Figure 5c, is the next most abundant ion fragment detected. Another fragment of levoglucosan, m/z 73 ($\mathrm{C_3H_5O_2}^+$) with a similar pattern is shown in Figure S2b. The median values of $f\mathrm{C_2H_4O_2}^+$ and $f\mathrm{C_3H_5O_2}^+$ range from 0.0013 to 0.051 and 0.0005 to 0.016, respectively (Table S1 and S2). The softwood Douglas fir is again distinct from the hardwoods with higher $f\mathrm{C_2H_4O_2}^+$ and more of a size dependence than in oak. The differences between wood types are an indicator that treating wood by its most abundant component, cellulose, is overly

simplistic and highlights the inhomogeneous nature of wood and the complexity of emission and secondary reaction or decomposition that follows direct emissions. Levoglucosan has the same chemical formula as the linked cellulose monomers. Polymer chain length of cellulose affects pyrolysis production of levoglucosan and other products (Mettler et al., 2012b), and small linkage changes to glucose monomers changes the reaction pathway balance between ring opening and levoglucosan production (Chen et al., 2016). Variations in polymer chain length could potentially explain differences in

emission of levoglucosan between wood types. $\mathrm{C_2H_4O_2}^+$ shows a mostly opposite pattern from other large molecules containing oxygen (not shown) which are a larger fraction in maple than in oak and much larger than in Douglas fir, and with an opposite temperature trend. This could be due to it being directly emitted instead of a reaction product.







**Figure 5. The fraction of CO⁺ (m/z 28, a), CHO⁺ (m/z 29, b), $C_2H_4O_2^+$ (m/z 60, c), $C_9H_7^+$ (m/z 115, d), $C_9H_{11}^+$ (m/z 119, e), $C_9H_{11}O_3^+$**
**(m/z 167, f) in total organics for maple, oak, Douglas fir, and cellulose ("C"). Each plot is ordered by fuel, then temperature, then**
**size. Markers are sized by wood size. Green markers are not pyrolyzed species. Bars of the box correspond to 25th and 75th**
**percentiles, and whiskers correspond to 10th and 90th percentiles.**





Aromatic species are commonly observed in burning emissions, generally associated with high temperature
reactions. M/z 115 ($C_9H_7^+$) is a marker for aromatics and is shown in Figure 6d. This fragment is more abundant, and the
abundance increases with temperature, in the hardwoods but not in Douglas fir. The peak $f\,C_9H_7^+$ at each temperature occurs
at the end of the experiment, when the highest temperatures are reached – at or just above the set point due to exothermic
reactions that heat the wood to beyond the set point (Park et al., 2010; Fawaz et al., 2020). This relationship between higher
temperatures and higher aromatic content could come from one of three sources. First, lignin is the only dominant fraction of
wood that has aromatic components, and it is released from wood at higher temperatures, so the $C_9H_7^+$ fragment could be
from an increase in direct lignin emission followed by fragmentation during the ionization process in the mass spectrometer.
Second, the lignin polymer could be emitted at a given temperature, but the $C_9H_7^+$ fragment could be from thermal
decomposition of the lignin, which breaks down into smaller compounds or fragments at higher temperatures, before further
fragmentation in the ionization process. Finally, the parent molecule of the $C_9H_7^+$ fragment could be produced by secondary
reaction inside or immediately outside of the wood. In combustion, aromatic formation is a function of flame temperature in
this temperature range (Lu et al., 2009).

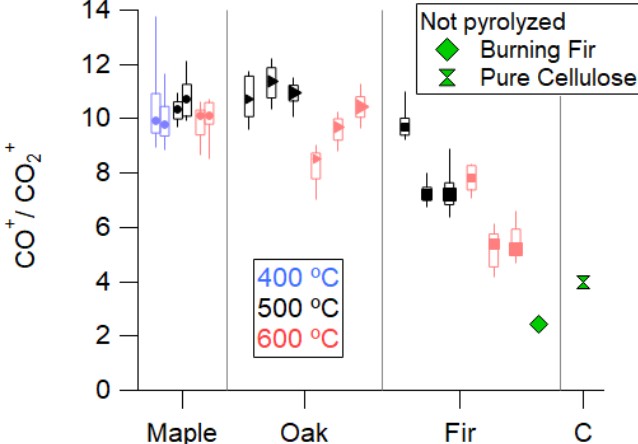

**Figure 6. The ratio of particle phase $CO^+$ to $CO_2^+$ fragments for maple, oak, Douglas fir, and cellulose ("C"). The order is by fuel,
then temperature, then size, markers at median are sized by wood size. Bars of the box correspond to 25th and 75th percentiles, and
whiskers correspond to 10th and 90th percentiles. Default AMS fragmentation table value is unity.**

A few individual fragments are unique to specific woods or conditions, as shown in Figure 5 e-f. $C_9H_{11}^+$ (m/z 119,
Figure 5e) is unique to Douglas fir wood, and more abundant at lower reactor temperatures, while $C_9H_{11}O_3^+$ (m/z 167, Figure
5f) is absent in Douglas fir, but also more abundant in cooler reactor temperatures for maple and oak. The inverse
relationship of these two fragments and wood type could indicate separate sources. The chemical formula $C_9H_{11}O_3^+$ indicates
an alcohol or ester substituted aromatic ring similar to that of lignin compounds. As such, it could be associated with





coniferyl or sinapyl alcohol. Sinapyl alcohol is a lignin (L3 in Figure 3) observed in hardwoods but not in softwoods, a trend that is reflected in this fragment as it is not observed in fir, the only softwood examined here. The other lignin types, coniferol and p-coumaryl, are present in all three woods. An analysis of the NIST spectra of individual, unpolymerized lignin components resulted in no fragments that are useful as markers for these compounds, although polymerization can further complicate the spectra.

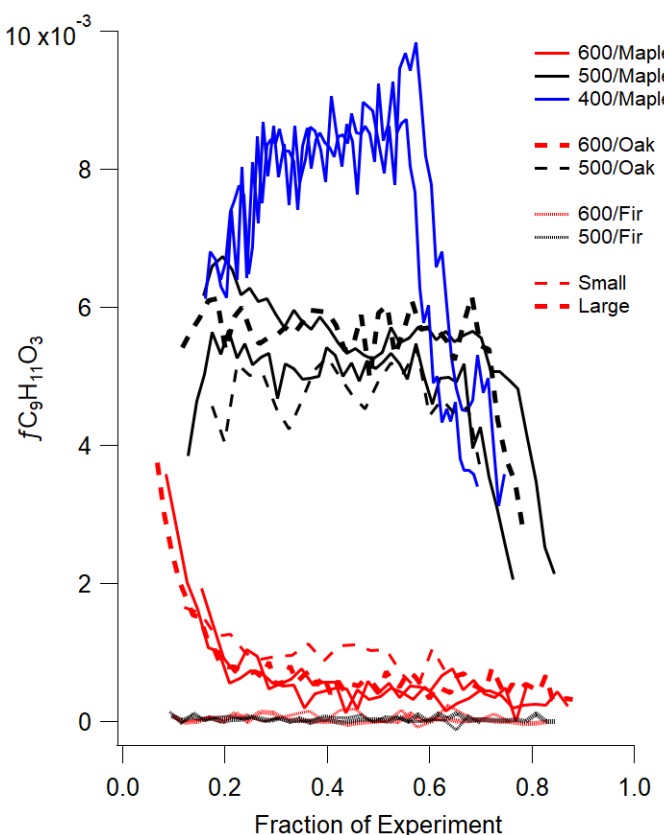

**Figure 7. The fraction of $C_9H_{11}O_3^+$ in the total organics for each small and large wood size experiment. The fragment is only in maple and oak spectra and is only emitted at the beginning of higher temperature set points before wood temperature has increased.**

In addition to fragments unique to specific wood types, some individual fragments show a temperature dependence when the fragment is plotted as a function of experimental time. For example, Figure 7 shows the fraction of the fragment $C_9H_{11}O_3^+$, which may be associated with coniferyl or sinapyl alcohol and is unique to the hardwoods. The emission pattern indicates that $C_9H_{11}O_3^+$ is most efficiently emitted around 400 °C. For 400 °C (blue lines), the fragment is relatively constant until near the end of the experiment, when $fC_9H_{11}O_3^+$ drops 60%, to the same range as the 500 °C experiments, before falling again to a lower fractional value. Either the fraction of wood at the appropriate temperature has decreased, or a secondary process (thermal decomposition or reaction) changes the parent compound. For 500 °C wood (black lines), the





fragment is less efficiently emitted, with a drop in fractional contribution at the end. At 600 °C (red lines), significant emission is only observed at the start of the experiment, where there is some lower-temperature emission. These trends are observed for maple and oak but not fir, which does not emit any of this fragment, or it has decomposed before detection. This association between an individual fragment, temperature, and a source provides insight into potential processes occurring during pyrolysis.

### 3.5. Pyrolysis in the context of open biomass and biofuel burning

In wild and controlled fires, pyrolysis occurs simultaneously with gas phase combustion, such that the aerosol emissions from open biomass or biofuel combustion are expected to contain a mixture of primary pyrolysis products and aerosol oxidized after emissions (i.e., in combustion). Pyrolysis aerosol emissions, as represented by the highly controlled, laboratory experiments conducted here on debarked, dried wood samples, are closely related to the underlying biomass chemical composition undergoing thermal decomposition, depolymerization, and fragmentation. Once flaming combustion of these emitted gas and particle products begins, the average oxidation state of the mixed pyrolysis and combustion aerosols increases in H/C and decreases in O/C (Sekimoto et al., 2018). The pyrolysis particle emissions measured here, which are similar to the wood component compositions and contain no refractory black carbon soot, are thus more oxygenated than fresh burning emissions, such as the laboratory burn of Douglas fir shown as a green diamond in Figure 3.

Once aerosols are emitted from open biomass/biofuel combustion, atmospheric chemical processes, including atmospheric oxidation and photolysis, further modify the chemical composition of these particles. These atmospheric aging processes increase O/C and decrease H/C ratios relative to fresh emissions. Figure 3 includes theoretical aging lines that correspond to reactions known to occur in the atmosphere, such as additions of carboxylic acid (slope=-1) and ketone or aldehyde (slope=-2) functional groups to a hydrocarbon backbone. Fresh biomass burning emissions, here represented by the Douglas fir burn (diamond marker), generally chemically transform along a slope between -1 and -2, effectively moving through the pyrolysis aerosol emission region of H/C vs O/C space. Thus, when included in a mixture of processes that includes combustion such as in wildfires, the presence of direct pyrolysis emissions may give the impression of atmospheric aging.

A distinct tracer for pyrolysis emissions are the anhydrosugars and their fragments (i.e., $f60$). These sugars may volatilize or be generated via thermal decomposition of woody components (i.e., cellulose). Figure S3 shows $f44$ vs $f60$ from these pyrolysis experiments in the context of ambient measurements of biomass burning aerosol where the solid lines encompass typical values of $f44$ vs $f60$ (Cubison et al. 2011). The measured values span most of the atmospherically observed $f60$ values, so it is possible that pyrolysis, rather than combustion, is responsible for most of the emitted levoglucosan detected during measurements of wood burning.

Another potential tracer for direct pyrolysis aerosol emissions, or at least an indicator of how much the pyrolysis direct emissions are modified by flaming combustion, is the presence of high $CO^+$ and the lack of $CO_2^+$ in aerosol mass spectra. The lack of $CO_2^+$, i.e., low $f44$, in pyrolysis emissions indicates that carboxylic acids are neither common in the





wood nor created through secondary chemical transformations. The average ratio of $CO^+$ to $CO_2^+$ ranged from 5.3 to 11.3 (see Figure 6). Previous work on primary emissions from heating stoves reported a ratio of 3.9 which decreased to 0.9-1.2 with oxidation (Corbin et al., 2015). This supports a hypothesis that primary heating stove emissions under cold-start or

addition of new fuel conditions are largely the result of pyrolysis. The $CO^+$ to $CO_2^+$ ratio is 2.44 for pure cellulose and 1.47 for the burning Douglas fir. This fragment ratio could be useful in determining the impact of pyrolysis when multiple burning processes are present.

## 4. Conclusions

Biomass burning emissions are a result of complex mixtures of pyrolysis and flaming and smoldering combustion

processes which occur across fuel types and burn conditions. Isolating atmospheric emissions from pyrolysis specifically has been challenging. This work measured the isolated, controlled open-reactor pyrolysis aerosol emissions of thermally thick wood with chemical composition aerosol instrumentation. Reactor temperature was the primary factor governing emitted particle loadings, where higher temperatures resulted in lower aerosol loadings. Directly emitted pyrolysis organic aerosol is more oxygenated than biomass burning emissions and contain sugar-like compounds. The chemical composition is distinctly

different from atmospherically oxidized hydrocarbons which contain carboxylic acid groups. Overall, the emissions are chemically similar between fuel types and reactor temperatures: $CO^+$ is the primary ion measured by the AMS, followed by other small, oxidized fragments and levoglucosan markers. Pyrolysis organic aerosol mass spectra are similar to previous studies of cookstoves (Corbin et al., 2015), especially in fuel-rich re-loading and pre-burning phases. A $CO^+/CO_2^+$ ratio greater than 5 is indicative of pyrolysis.

Several individual ions are unique to an individual wood or pyrolysis temperature and can be used as indicators of that wood type. $C_9H_{11}^+$ (m/z 119) is unique to low-temperature fir while $C_9H_{11}O_3^+$ (m/z 167) is unique to the low-temperature hardwoods maple and oak. The differences in emission between temperatures are the result of complex ejection and secondary reaction mechanisms that cannot be de-coupled here. Future studies of slower or stepwise temperature ramping are required to examine more closely whether temperature effects reflect primary ejection or secondary reaction.

The temperature dependent nature of some fragments is important context for larger mixed-fuel wildfires where a broader range of temperatures exist, in addition to the gas phase combustion of emitted products.

This work suggests that pyrolysis contributes meaningfully to organic aerosol emissions. Pyrolysis products are not the only emission from combustion or wildfires, as gas-phase reactions consume the products that are directly emitted and create new products, and smoldering reactions at the wood surface may emit yet a different set of products. Placing these

pyrolysis results in the wider systems of combustion and wildfires is challenging due to flaming and smoldering reactions which consume the direct pyrolysis products analyzed here. Future work analyzing gas phase emissions as well as the volatility distribution of pyrolysis products can bring chemical closure to this system.



**Acknowledgements:**

The authors would like to thank Indoor Climate Research and Training for hosting the experiments. This research was
funded by the National Science Foundation under grant AGS-1742956. We also gratefully acknowledge Brian Heffernan for
his help in the dry generation of cellulose experiment.

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
