# Peer review of "Chemically distinct particle phase emissions from highly controlled pyrolysis of three wood types"

_Atmospheric Chemistry and Physics, 2022_

## Author Comment (AC1)

**Comment on acp-2022-535**

Anonymous Referee #1

**General comments**

This paper by Avery et al. attempts to understand the contribution that pyrolysis makes to the emissions from biomass burning in the form of aerosol production form different wood types. This is important because pyrolysis underpins the processes of combustion and understanding the yield and composition of aerosols from is important for assessing impacts on climate and human health. This paper examines typical fragments in mass spectra generated by an aerosol mass spectrometer to detail the patterns in terms of yields and emission factors, and tries to explain the processes occurring that make up the aerosols that are measured. It builds upon the experimental work conducted by Fawaz et al. (2021) who characterise the pyrolysis chamber used in the experiment to answer questions as those posed in this paper.

Although the paper attempts to address relevant questions, and produces some interesting and new insights, I find its conclusions weak based up important information missing from the manuscript. Most importantly, is the number of replicate runs done in this study. In lines, 169-171 it mentions running one wood type in duplicate only. There is no mention as to how many replicates of the other two woods were examined at the different temperatures and sizes. Indeed, in Table 1, which the authors state in L179 is a full list of experimental conditions, the duplicate results are shown for the one wood type but the other woods and conditions are only shown as one replicate. Also, several of the graphs only show one line for some of the experimental conditions but two for the material that was stated to be run in duplicate. If only one replicate was run, then the lack of replication in the experiment does need not give me confidence in the findings of this study and undermines statements made by the authors about high reproducibility.

The authors thank the reviewer for a thorough review.

The manuscript accurately reflects the number of replicates analyzed in this work. However, we demonstrated and published the high degree of repeatability of the particle concentrations emitted from pyrolysis experiments conducted using our pyrolysis reactor in Fawaz et al. (2021), which showed the repeatability in great detail. The current manuscript expands upon our previous work by focusing on identifying chemical markers in the pyrolysis particle emissions as a function of differences in pyrolysis conditions.  For this work, we ran duplicate experiments on small maple wood samples at three different reactor temperatures, showing repeatability in emitted particle concentrations comparable to Fawaz et al. (2021).  Given our ability to reproduce consistent particles emissions at a different time from the same reactor, we used our remaining study time and resources to sample a range of unique wood types, sizes, and reactor temperatures.

We have modified the text to include the following: L74: "A related paper describes yields and product distributions of pyrolysis and demonstrates the high repeatability of gas and particle phase emissions using the same reactor over a larger range of experimental conditions than described here (Fawaz et al., 2021)."

L179: "Maple experiments were performed in duplicate on only small wood at all three temperatures. These duplicates reflected the high level of experimental repeatability shown in Fawaz et al. (2021), so replicates were not sampled for other pyrolysis conditions. Douglas fir and oak were pyrolyzed at only 500 ºC and 600 ºC but at all three sizes."

**Specific comments**

L71-76: This is your aim of the study. I suggest moving this to the end of the introduction

We agree that this statement summarizes the aim of our study. We have chosen to place it in the initial introductory section to make our intent clear. We then use the remaining subsections of the introduction to add more context to our. We have added the following endings to each of the introduction sections to make clear our organization:

Section 1.1 ends with (L79):

"Before describing the methods, we discuss other lines of investigations into biomass decomposition products to place our work in context."

Section 1.2 ends with (L116):

"This work specifically isolates emissions from the pyrolysis phase."

Section 1.4 ends with: (L168):

"This work examines decomposition products of wood under realistic pyrolysis conditions."

L101-103: What is the relevance of mentioning PMF? It's not a technique you use in this study.

PMF is a common way of identifying sources of ambient aerosol, especially biomass burning emissions. PMF was not used in this study, but our resulting aerosol mass spectra may be useful for understand ambient BBOA source spectra obtained through PMF analysis. For example, the value of f60 was observed to be similar between our laboratory pyrolysis emissions and ambient biomass burning emissions from literature, consistent with pyrolysis as a potential source of the ambient f60 BBOA marker from wildland fires.

The text has been modified to include this point:

"Positive Matrix Factorization (PMF) applied to AMS mass spectral data (Paatero and Tapper, 1994; Ulbrich et al., 2009) has identified several typical mass spectral signatures for biomass burning organic aerosol (BBOA), usually distinguished from other factors by significant contribution at m/z 60."

L104-109: This reads as justification for methodology. I'd suggest moving this to the methods section.

These lines were included as part of a review of prior literature and indicate that different types of "biomass burning" have different chemical characteristics.

The text has been modified to clarify this point:

"Other sources of m/z 60 usually attributed as "biomass burning", including residential heating and cooking stoves, exhibit some similar characteristics to uncontrolled burning in m/z 60 contribution, but have some differences."

L127-128: What is the relevance of this statement? You don't do modelling in this paper.

The modeling work of Fawaz et al. (2020) found that known fundamental processes can explain the mass emissions profiles observed in Fawaz et al. (2021) and this work. This finding supports the statement that controlled pyrolysis is reproducible and explainable.

The text has been modified to make this link:

"Fawaz et al. (2020) used thermal diffusivity and permeability to model open pyrolysis emissions in the particle and gas phases, and found these processes explain the mass emission profiles described here."

L134-137: Like L71-76, this is part of your aims of the study. I'd suggest moving it to the end of the introduction.

Because of the very different type of investigations into biomass emissions, we have chosen an organization in which each type of exploration is summarized, and then or contribution is placed in that context.

L151-152: Missing reference for the pyrolysis temperatures of cellulose and hemi- cellulose.

Yang et al. (2007) is the reference for all 3 wood components. The reference has been moved to the previous sentence so as to read:

"The primary components of wood decompose from the polymer matrix and are emitted at different temperatures (Yang et al., 2007). Hemicellulose decomposes at 200-300 °C, followed by cellulose between 300-400 °C. Lignin is emitted across a broad range of temperatures spanning the ranges of both hemicellulose and cellulose, and up to 900 °C."

Section 2.1: The title for this section needs to include material characterisation. All, I would expect some basic characterisation of the wood to have been presented. For example, density, proximate or ultimate analysis, cellulose, hemi-cellulose, and lignin contents. The compositional data would be important for explaining f60 and f73 results, for example.

Section 2.1 title now reads: "Pyrolysis reactor and material characterization".

The density of the wood samples, described in section 3.2, have been added to section 2.1.

We were unable to obtain analyses of the specific cellulose, hemicellulose, and lignin contents for these samples during the study. Such analyses might help inform the differences in chemistry observed between the wood types. However, they are frequently not reported in literature on pyrolysis, even those that examine composition of liquid products (e.g. Branca et al 2003). This common omission from common pyrolysis studies may occur because reactions occur between the components, as we also suspect. Furthermore, the link to f60 and f73 are for comparison with a broad range of wood burning emissions in ambient air; thus, the exact composition of pyrolyzed wood here is not important.

Section 2.1 now reads:

"Three species of wood were pyrolyzed: white oak (*Quercus alba*), hard maple (*Acer nigrum*), and Douglas fir (*Pseudotsuga menziesii*). The apparent densities of these woods are 860±27, 750±17, and 473±28 kg/m$^3$, respectively."

L167-168: The Genus (in Genus species) should start with a capital letter

Thank you for this correction; it has been changed as shown above.

L176-179: You talk about changing the dilution ratio but don't describe how (either here or in the supplemental material). This is important to know in order for this experiment to be replicated.

The dilution ratio was measured before and after each experiment by measuring the exhaust and sample flows. The text has been modified to clarify the measurement:

"Aerosol loading values reported here are as sampled from the exhaust duct, only accounting for the secondary dilution. The dilution ratio was set by controlling the mount of dilution relative to exhaust duct air. The sample flow, or the difference between exhaust and dilution flow, was measured before and after each experiment, and the dilution ratio (DR) is the ratio between the exhaust flow and sample flow. The dilution setup and measurement technique were changed between the maple and other wood samples. For maple, the dilution ratio is an estimate based on the aerosol mass loading compared with other woods. For oak and Douglas fir, the DR was measured directly. A full list of experimental conditions including DR are shown in Table 1."

L191: Who is the developer of the Squirrel and Pika? Please insert details.

The text has been modified to read:

Data were analyzed with standard Squirrel and Pika (version 1.61F and 1.21F, respectively) packages for Igor Pro software (Sueper et al., 2023).

And the following citation has been added to the references list:

D. Sueper and collaborators, ToF-AMS Data Analysis Software Webpage, CU Boulder
http://cires1.colorado.edu/jimenez-group/wiki/index.php/ToF-AMS_Analysis_Software, 2023.

Table 1: Description of abbreviations is not given in the legend. Also, the asterisk seems to have been used for two different purposes. These needs changing for one use and the other use described with the appropriate symbol. Also, what do the plus/minus values represent? Standard deviations, ranges?

Table 1 has been modified to include the requested clarifications.

**Table 1. Summary of experimental conditions and results of chemical properties and emission-related ratios of wood pyrolysis products, including temperature, size, wood, dilution ratio (DR), average oxidation state (OsC) ± standard deviation, ratio of $CO^+$ and $CO_2^+$ fragments ($CO^+/CO_2^+$), emission ratio (ER), emission index (EI), and total loading. The DR for maple is estimated based on aerosol emissions at similar conditions to oak and fir, when the DRs were measured directly.**

| Temp (°C) | Size | Wood | DR | OsC Avg | $CO^+/CO_2^+$ ratio | ER ($\mu g/m^3$/ppm)* | EI (g/g)* | Total loading (g)* |
|---|---|---|---|---|---|---|---|---|
| 400 | S | Maple | 50† | -0.30 ± 0.06 | 9.9 | 8300 | 0.80 | 11 |
| 400 | S | Maple | 50† | -0.28 ± 0.08 | 9.8 | 8300 | 0.80 | 10 |
| 500 | S | Maple | 50† | -0.29 ± 0.09 | 10 | 3800 | 0.48 | 8.4 |
| 500 | S | Maple | 50† | -0.27 ± 0.06 | 11 | 3800 | 0.52 | 8.2 |
| 600 | S | Maple | 50† | 0.09 ± 0.03 | 10 | 140 | 0.06 | 1.2 |
| 600 | S | Maple | 50† | 0.06 ± 0.03 | 10 | 190 | 0.07 | 1.4 |
| 500 | S | Oak | 190 | 0.01 ± 0.02 | 11 | 1600 | 0.23 | 3.4 |
| 500 | M | Oak | 211 | -0.02 ± 0.03 | 11 | 1800 | 0.29 | 7.2 |
| 500 | L | Oak | 316 | -0.18 ± 0.06 | 11 | 2500 | 0.39 | 22 |
| 600 | S | Oak | 203 | 0.13 ± 0.04 | 8.5 | 270 | 0.10 | 1.4 |
| 600 | M | Oak | 190 | 0.07 ± 0.03 | 9.7 | 380 | 0.08 | 3.3 |
| 600 | L | Oak | 203 | -0.11 ± 0.05 | 10 | 540 | 0.09 | 8.5 |
| 500 | S | Fir | 211 | -0.07 ± 0.10 | 9.7 | 8400 | 0.71 | 6.3 |
| 500 | M | Fir | 193 | -0.17 ± 0.06 | 7.2 | 3100 | 0.31 | 7.3 |
| 500 | L | Fir | 277 | -0.25 ± 0.07 | 7.2 | 5400 | 0.51 | 23 |
| 600 | S | Fir | 203 | 0.07 ± 0.06 | 7.8 | 960 | 0.19 | 2.2 |
| 600 | M | Fir | 238 | -0.12 ± 0.06 | 5.4 | 700 | 0.13 | 3.7 |
| 600 | L | Fir | 492 | -0.17 ± 0.07 | 5.2 | 1200 | 0.25 | 16 |

* Dilution ratio applied.

†Dilution ratio estimated rather than measured directly.

L274: Reproducibility should be repeatability – you are using the same equipment, materials etc., all within the same timeframe. However, what metric are you using to demonstrate this? You don't present one.

Reproducibility has been replaced with repeatability. Replicates of this experimental procedure are described in Fawaz et al. (2021). The text now reads:

"The high repeatability of these temporal profiles, as demonstrated in Fawaz et al. (2021) and reproduced for this study using small maple samples at each temperature, enables estimates of open-reactor pyrolysis emission-related enhancement ratios (ER; OA/CO ratio in $\mu g/m^3$/ppm)"

L278-279: How did you calculate your uncertainties? A comment in the methods section is needed to explain this.

As noted in the text, "The ER values are the fitted slope of measured organic aerosol ($\mu g/m3$) times the secondary dilution ratio (DR) versus the measured CO (ppm)."

The uncertainty estimates for ER values were obtained by summing the square of individual measurement uncertainty estimates.  The AMS mass measurement uncertainty was estimated at ±38% (Bahreini et al., 2009), the CO concentration at ±10% and secondary dilution at ±10% when measured, and ±100% when estimated, giving an estimate uncertainty of the calculated ER's of ±40% (107%).

L321 – 323: You need a reference to the FIREX campaign.

A reference to the FIREX campaign of Selimovic et al., 2018 has been added.

L443 – 446: How did you make this connection? Having characteristics of the wood would substantiate this comment.

This comment highlights that the range in f60 is large but that the source of that diversity could come from several sources including secondary reactions. We do not have access to specific wood characteristics and note that the range of f60 values here is representative of fresh emissions as described in Cubison et al. (2011).

The text has been edited as the following:

"The differences between wood types are an indicator that treating wood by any single component or marker, is overly simplistic and highlights the inhomogeneous nature of wood and the complexity of emission and secondary reaction or decomposition that follows direct emissions."

L520 – 522: What is the implication of this finding?

In Van Krevelen diagrams, pyrolysis emissions lie in the same chemical (i.e., O:C and H:C) space as aged aerosols, though the oxygen-containing functional groups differ significantly, as illustrated by the lack of substantial f44 signal (see Figure 8). Therefore, care should be taken to use additional markers to identify the age of a biomass burning plume.

The following text has been added:

"Therefore, additional markers and indicators beyond O:C and H:C ratios should be used to identify the age and presence of pyrolysis emissions in a plume."

L523 – 528: If this is important to show why is the graph in the supplementary materials?

This figure has been moved from SI to become Figure 8 with the same caption.

L553 – 556: What you are suggesting is coupling the AMS to thermogravimetric analysis (TGA) which is a common technique for pyrolysis studies. There are an abundance of studies on TGA and gas analysis which you could cite, but not on aerosols and their composition.

We agree that there are many studies of pyrolysis that report the product gases, but not the aerosol composition. In studies of industrial pyrolysis, emitted material that condenses is typically lumped into the category "tar". In addition, industrial work usually examines fast pyrolysis that occurs in thermally-thin material where the rate of heat transfer does not affect reaction rates.

 We have modified the text as follows:

"Future studies of aerosol composition as the result of slower or stepwise temperature ramping are required to examine more closely whether temperature effects reflect primary ejection or secondary reaction. These investigations would differ from those reported in industrial pyrolysis studies because they would examine particle composition rather than gas composition, and they would examine emissions from thermally thick biomass."

Figure S3: There is no explanation in the legend of the dashed line at 0.3% being the background f60 as described by Cubison et al. (2011). Nor is there an explanation that the solid lines are the expected relevant ratios for the atmosphere.

The caption has been modified as follows, and is now Figure 8:

"**Figure 8. Fraction of $CO_2^+$ (m/z 44) to fraction of $C_2H_4O_2^+$ (m/z 60) as an indicator of atmospherically relevant biomass burning. Each value is an experiment average with bars indicating the standard deviation. The dotted vertical line represents the nominal ambient background value of 0.3% from Cubison et al. (2011). The solid lines forming two-thirds of a triangle represent the bounds observed by Cubison et al. (2011), which would be generally expected for measurements of fresh to aged biomass burning in the atmosphere.**"

**Technical comments:**

All tables: Does not match the formatting requirements for the Journal i.e your use of vertical lines.

I believe the reviewer is referring to the horizontal lines separating the experiment types by wood and reactor temperature, as there are no guidelines on vertical lines. We note that the guidelines state that horizontal lines "should normally only appear above and below the table" but that there are many exceptions to this to guide the eye including the tables in Fawaz et al., 2021 (ACP). We hope that we can come to a satisfactory exception or compromise with the editor and typesetter for final publication.

All tables and graph: What are the number of independent samples?

All results presented are from independent samples.

**Comment on acp-2022-535**

Anonymous Referee #2

The paper titled "Chemically distinct particle-phase emissions from highly controlled pyrolysis of three wood types", by Avery and Co-Authors presents the results of 18 experiments in which the authors investigate the chemical composition of aerosols generated during the pyrolysis of wood typical of the western US. The authors investigate 3 variables: wood type, wood size, and temperature. The results are corroborated by the comparison with the chemical composition of aerosolized cellulose carried out in the lab. One of the major strengths of the results presented here is that the experimental setup guarantees that the aerosols are generated by pyrolysis only, with no combustion happening, thanks to the use of a heated nitrogen flow. The study is of interest to the scientific community and addresses relevant scientific questions, such as the chemical composition of aerosol generated during pyrolysis. This topic is well within the scope of ACP.

The main weaknesses of the paper are the lack of a schematic picture of the pyrolysis reactor, the relatively small number of repeated experiments, the lack of explanation of the uncertainties, and the lack of measured dilution for the maple samples.

The data presented are novel and the conclusions reached are substantial and will be of use for the interpretation of future data (e.g., the oxidation state of emitted OA that remains largely constant over the course of a pyrolysis experiment, and that $CO^+/CO_2^+$ greater than five is identified as a marker for pyrolysis). The methods are valid, in particular, the repeatability of the temporal profiles is remarkable.

Title: I recommend either adding a hyphen to "particle-phase" or just using "particles"

A hyphen has been added to the title to read particle-phase

40 biofuel burning, I recommend adding a reference

The sentence provides a description of the difference between biofuel and biomass burning, and is intended only to illustrate some of the uses of fuel (deliberate, controlled use) compared with wildfires."

84 "Previous work has shown …", citation needed

The following citation has been added:

S.K. Akagi, R.J. Yokelson, C. Wiedinmyer, M.J. Alvarado, J.S. Reid, T. Karl, J.D. Crounse, P.O. Wennberg, Emission factors for open and domestic biomass burning for use in atmospheric models, Atmospheric Chemistry and Physics. 11 (2011) 4039–4072. https://doi.org/10.5194/acp-11-4039-2011.

135 please add which different conditions are explored

This sentence has been changed to the following:

"Its purpose is to assess how the pyrolysis process under different conditions of wood type, pyrolysis temperature, and wood size, may contribute to biomass burning emissions."

160 Section 2.1, I recommend adding a schematic picture of the reactor it would help the reader follow the text without having to dig up the previous paper cited.

We understand the reviewer's request; however, given space constraints and the significant connections between this manuscript and the recently published Fawaz et al. (2021), including the use of the same pyrolysis reactor – detailed in Fawaz et al. (2021) – and the importance of the repeatability results obtained in Fawaz et al. (2021), we have decided to not include a schematic figure in this manuscript.

166 please add that the wood was cut along the grain lengthwise as it's important later on in the paper

The text has been modified to read:

"Pieces were cut along the grain lengthwise to small (3.5 x 3.8 x 2.9 cm), medium (7 x 3.8 x 2.9 cm), or large (14 x 3.8 x 2.9 cm) sizes."

173 $m^3$/s correct the format for the units

The text has been modified with the appropriate superscript.

175 "The aerosol sampling line was further diluted" Please add the range of dilutions used

The dilution ratios are given in Table 1 and ranged between 50 and 500. This sentence has been modified to read:

"The aerosol sampling line was further diluted with filtered compressed air by a factor ranging from 50 to 500 (see Table 1)."

187 "The size distribution of particles was well within the AMS standard lens size range" I recommend corroborating this statement with a picture in the SI

The following figure has been added to the SI, and the following text has been added:

Figure S1 shows an example size distribution indicating the emitted particles fall within this range.

[Figure]

Figure S1. An example size distribution from Fir pyrolyzed at 600°C, size large wood.

278 "The uncertainty for the ER ..." add here or in the method section a line/citation on how they are calculated

As noted in the text, "The ER values are the fitted slope of measured organic aerosol (µg/m3) times the secondary dilution ratio (DR) versus the measured CO (ppm)."

The uncertainty estimates for ER values were obtained by summing the square of individual measurement uncertainty estimates.  The AMS mass measurement uncertainty was estimated at 38% (Bahreini et al., 2009), the CO concentration at 10% and secondary dilution at 10% when measured, and 100% when estimated, giving an estimate uncertainty of the calculated ER's of 40% (107%).

288 "The uncertainty for the EI ..." add here or in the method section a line/citation on how they are calculated

As noted in the text, "The emission index of particle phase organic mass is presented in Figure 2d-f as the measured organic aerosol emission rate (g/s), that is the measured OA (µg/m3) times DR times flow rate through the duct (m3/s), as a function of wood mass loss rate (g/s). The slope gives mass-based EI values. The uncertainty of the EI values is greater than the ER values due to the assumption that all emissions enter the exhaust duct."

Thus, we did not provide actual uncertainty estimates for EI values, stating that they are greater than the ER uncertainties.